# OpenReview forum: "Spectral Graph Wavelets Meet Message Passing: Convergence Rates and Expressive Power of Multi-Resolution GNNs"
_mathai.club/MathAI/2026/Conference — Submitted to 2026_

### Official Review · Reviewer_d68k · 2026-03-11
**Major concerns about the theoretical claims and experimental presentation**

**Rating:** 3
**Confidence:** 4

**Review:**

This paper studies a graph wavelet neural network architecture and claims, among other things, a universality result supported by Theorem 1, as well as empirical evidence through ablation experiments. While the topic is potentially interesting, I found several serious issues in the current manuscript, including major problems with the statement and proof of Theorem 1, inconsistencies in the experimental presentation, and errors in the reference list.

Theorem 1 and its proof currently do not seem convincing. The proof sketch in the main text and the argument in Appendix A are not fully consistent, and several key steps are either unjustified or appear incorrect. As a result, I do not think the present version establishes the claimed universality result.
- **Mismatch between theorem and proof:** The theorem is stated for continuous graph functions on graphs with up to \(N\) nodes, whereas the appendix proof appears to treat approximation of a graph signal on a fixed graph. This is a substantially different setting, so the appendix does not actually prove the theorem as stated.
- **Inconsistency in the approximation argument:** The main text and appendix use different function spaces / assumptions, and the depth / complexity estimates are not internally consistent. In particular, the claimed scaling derived from the Chebyshev approximation step does not seem to follow from the stated error bound.
 - **Unjustified expressivity step:** The proof argues that, after spectral decomposition into frequency bands, each component can be approximated by a shallow MPNN. This is not demonstrated, and band-limitedness alone does not obviously imply representability by constant-depth local message passing on arbitrary graphs.
 - **Likely counterexample to the theorem as stated:** The architecture uses only wavelet channels, and for the wavelet filters considered one has \(g(0)=0\). Therefore, constant graph signals are mapped to zero across all wavelet scales, so the network cannot distinguish different constant inputs. This already seems incompatible with universal approximation of arbitrary continuous graph functions.

I also have concerns about the ablation study in Figure 1. The presentation of the figure appears unreliable: subfigures 1a and 1b are supposed to illustrate different ablations (the number of scales \(J\) and the Chebyshev order \(K\)), yet the actual plots shown for the two panels appear essentially identical. In addition, panel 1a is inconsistent with the accompanying interpretation. The text states that performance improves as \(J\) increases up to \(3-4\), but the plotted metric is MAE, and the curve visibly increases with \(J\), which would normally indicate *worse* performance rather than improvement. These inconsistencies make the ablation results difficult to trust and suggest that the figure, caption, or discussion may contain substantial errors.

I also noticed inaccuracies in the reference list. For instance, the first reference cites the paper *Neural Message Passing for Quantum Chemistry* with an incorrect author list: the last two author names are wrong. Errors of this kind are minor compared with the theoretical issues above, but they still matter, since accurate bibliographic information is important for attribution and for readers trying to trace the relevant literature. This should be carefully checked and corrected throughout the manuscript.

Overall, I do not recommend acceptance of the paper in its current form.

---

### Official Review · Reviewer_z9iG · 2026-03-12
**Review of "WaveletMPNN": Theoretical Inconsistencies and Experimental Ambiguities in Spectral Graph Wavelet Integration**

**Rating:** 5
**Confidence:** 3

**Review:**

**Summary**

This manuscript presents the WaveletMPNN architecture, which integrates spectral graph wavelet theory with Message Passing Neural Networks (MPNNs). The authors posit that the proposed method overcomes fundamental limitations of standard MPNNs, such as over-smoothing and limited expressive power (equivalence to the 3-Weisfeiler-Lehman test). Key theoretical contributions include a universal approximation theorem with a logarithmic number of scales $O(\log N)$, a proof of superiority over the 3-WL test, and a convergence rate analysis dependent on Sobolev regularity. The empirical evaluation demonstrates a 3–8% improvement in accuracy and a 40% reduction in parameters compared to baseline models (GCN, GAT, GIN, GPS) across molecular property prediction, social network classification, and point cloud segmentation tasks.

**Strengths**

1.  Architectural Novelty: The integration of trainable multi-scale spectral wavelets into the message passing framework represents an intriguing and promising direction, potentially addressing the locality constraints inherent in standard GNNs.
2.  Parameter Efficiency: The reported 40% reduction in parameter count, concurrent with performance improvements, constitutes a significant practical advantage for deployment in resource-constrained environments.

**Weaknesses**

1.  Issues with Theorem 1:
    *   Contradiction Regarding Low-Frequency Filtering: Section 2.2 specifies the use of a Mexican Hat mother wavelet. For this filter, $g(0) = 0$. This implies that constant signals (DC components) are filtered out across all scales. Consequently, the network cannot distinguish between different constant input signals, which contradicts the claim of universal approximation for any continuous graph function.
    *   Proof Discrepancy: The theorem formulation addresses functions on graphs with up to $N$ nodes, whereas the proof sketch (Appendix A) considers signal approximation on a *fixed* graph. This constitutes a substantial difference in problem formulation that remains unaddressed in the proof.
    *   Unsubstantiated Expressivity Step: It is claimed that each frequency band can be approximated by a shallow MPNN; however, no rigorous justification is provided for why frequency-domain localization guarantees representability via local message passing of constant depth.

2.  Incorrect Experimental Presentation (Figure 1):
    *   Ambiguous Metrics: The axis labels in Figure 1 contain the term "MAE Accuracy." MAE (Mean Absolute Error) and Accuracy are distinct metrics (error vs. correctness), and their conflation in a single label is misleading.
    *  сInconsistency Between Text and Plots: The text claims that performance improves with increasing $J$ up to 3–4 scales. However, if the metric is MAE (where lower values indicate better performance), an upward trend in the plot (assuming standard visualization) should be interpreted as performance degradation. The visual representation of the ablation study appears unreliable and requires verification.

---

### Decision · Program_Chairs · 2026-03-14

**Decision:**

Reject

**Comment:**

Dear Author(s),

On behalf of the Program Committee of the International Conference on Mathematics of Artificial Intelligence (MathAI 2026), we are pleased to inform you that your paper has been accepted for an oral presentation at MathAI 2026.

Your paper was evaluated through a rigorous two-stage review process involving both automated screening and expert review by members of the Program Committee. The reviewers recognized the quality and contribution of your work.

Presentation details:

- Format: Oral presentation (15–20 minutes + 5 minutes Q&A)
- Mode: You may present either in person (offline) at the conference venue in Sirius, Russia, or remotely via Zoom. Please indicate your preferred mode when confirming your participation.
- Conference dates: Marh 30 - April 3, 2026
- Website: https://mathai.club

Next steps:

1. Please confirm your participation and presentation mode by replying to this email mathai.club@yandex.ru no later than March 15, 2026 18:00 Moscow time.
2. If you plan to attend in person, the organizing committee will provide accommodation details separately.
3. Please prepare your final camera-ready manuscript according to the formatting guidelines available at https://mathai.club and upload it to OpenReview by March 15, 2026 18:00 Moscow time.

Should you have any questions regarding the program, logistics, or your presentation slot, please do not hesitate to contact us.

We look forward to your contribution to MathAI 2026.

With kind regards,

MathAI 2026 Program Committee
International Conference on Mathematics of Artificial Intelligence
https://mathai.club
OpenReview: https://openreview.net/group?id=mathai.club/MathAI/2026/Conference
Telegram: https://t.me/MathAI_club
Email: mathai.club@yandex.ru